# Safety and Efficacy of Post-Eradication Smallpox Vaccine as an Mpox Vaccine: A Systematic Review with Meta-Analysis

**DOI:** 10.3390/ijerph20042963

**Published:** 2023-02-08

**Authors:** Shelia M. Malone, Amal K. Mitra, Nwanne A. Onumah, Alexis Brown, Lena M. Jones, Da’Chirion Tresvant, Cagney S. Brown, Austine U. Onyia, Faith O. Iseguede

**Affiliations:** Department of Epidemiology and Biostatistics, School of Public Health, College of Health Sciences, Jackson State University, Jackson, MS 39217, USA

**Keywords:** vaccine, monkeypox, smallpox, vaccinia, safety, efficacy

## Abstract

According to the World Health Organization, 83,339 laboratory-confirmed cases, including 72 deaths, of mpox (formerly known as monkeypox), have been reported from 110 locations globally as of 20 December 2022, making the disease a public health concern. Most of the cases (56,171, 67.4%) were reported from countries in North America. Limited data on vaccine effectiveness in the current mpox outbreak are available. However, the modified vaccinia virus (smallpox vaccine) has been predicted to prevent or reduce the severity of the mpox infection. The present study of systematic review and meta-analysis aimed to evaluate the modified vaccinia vaccine’s safety and efficacy on mpox by using reported randomized clinical trials. Following guidelines from the Cochrane Collaboration and PRISMA, multiple databases including PubMed, PLOS ONE, Google Scholar, British Medical Journal, and the U. S. National Library of Medicine were searched. Out of 13,294 research articles initially identified, 187 were screened after removing duplicates. Following the inclusion and exclusion criteria, the meta-analysis included ten studies with 7430 patients. Three researchers independently assessed the risk of bias in the included study. The pooled results suggest that the vaccinia-exposed group had fewer side effects when compared to the vaccinia naïve group (odds ratio: 1.66; 95% CI: 1.07–2.57; *p* = 0.03). Overall, the modified vaccinia has proven safe and effective in both vaccinia naïve and previously exposed groups, with higher efficacy in the previously exposed groups.

## 1. Introduction

Mpox (formerly known as monkeypox) is caused by monkeypox virus (MPX), which is part of the same family of viruses as variola virus that caused smallpox. As of December 20, 2022, a total of 83,339 laboratory-confirmed cases, 1532 probable cases, and 72 deaths have been reported from 110 countries [1]. One species of the genus *Orthopoxvirus* in the family *Poxviridae* is responsible for the MPX virus, which is a zoonotic pathogen in humans. MPX is the most frequent systemic infection [2], which occurs both within and outside of endemic countries in travelers returning from western and central Africa [3]. The MPX outbreak in 2003 in the United States was linked to infected animals imported from elsewhere [3]. The virus was then considered an emerging infection and a potential public health threat. Unfortunately, it has been confirmed in subsequent MPX outbreaks [3], including the current ongoing outbreak. The increased incidence of mpox in nonendemic areas, and the emergence of potentially weaponizable virulent strains, are causes to raise public health awareness.

New safe and efficacious vaccines are needed for mpox. Vaccinia, the vaccine used for smallpox eradication, has attracted renewed interest in the prevention of mpox. Vaccinia was safe and effective for most people [4]. Through several observational studies, vaccinia demonstrated approximately 85% effectiveness in preventing mpox [1].

A study by Xuan et al. (2022) [2] demonstrated pathophysiology of mpox infection. The researchers included mpox virus-infected in vitro models as subjects to distinguish between cells actively dividing and reacting in conjunction with genes regulated by a common or standard factor during in-trial mpox disease activity. Among multiple immune pathways of interest, the researchers also found significantly expressed genes that played pivotal roles during mpox infection. Their findings might contribute to specific mechanisms of the infection, post-infection complications or conditions, underlying immune response, and vaccine propagation.

Ongoing clinical trials evaluate the safety and immunogenicity of modified vaccinia as a viral vector for recombinant vaccines, immunotherapies, and oncolytic therapies to combat mpox [5]. There is little doubt that recombinant vaccines can be useful, but data need to be more comprehensive. How well these vaccines protect against mpox and the vaccination’s length of immunity is unknown. The amount of variability in protection by giving a single dose versus two doses is also unknown. The efficacy of the vaccine among people after exposure is still being studied [6]. This systematic review and meta-analysis aimed to examine the safety and efficacy of modified vaccinia for mpox.

## 2. Materials and Methods

This systematic review and meta-analysis were performed in accordance with guidance from the Cochrane Collaboration that included the international standards for conducting and reporting systematic reviews [7] and the Preferred Reporting Items for Systematic Reviews and Meta-Analyses (PRISMA) [8].

### 2.1. Search Strategy

The searches included MEDLINE (via PubMed), PLOS ONE, Google Scholar, British Medical Journal (BMJ), and the U.S. National Library of Medicine (accessed using www.clinicaltrials.gov). All published mpox and vaccinia literature reported from 1 January 1997, through 7 September 2022 (the last search date) was examined for eligibility. In PubMed, the searches included Medical Subject Headings (MeSH) and were limited to titles and abstracts. The search string used in PubMed was Monkeypox[MeSH] OR “Monkeypox virus”[MeSH] OR monkeypox[tiab] OR “monkeypox” OR “variole du singe” OR “variole simienne” and Smallpox[MeSH] OR “Smallpox virus”[MeSH] OR Smallpox OR “ Smallpox “ OR “vaccinia”, and in PLOS ONE was *monkeypox* *clinical trial* *OR* *odds ratio* *CI* *confidence interval* and intext: *CI* *OR* *Monkeypox vaccine*. In Google Scholar and Google, additional searches were executed using the following terms: monkeypox, Intext: Smallpox, and vaccinia and monkeypox. The search criteria for clinicaltrials.gov was Condition or disease = Smallpox, vaccinia, and other terms = vaccine, clinical trial. Only clinical trials that had results reported were included.

### 2.2. Data Extraction

After the article collection from the four databases, the duplicates were removed, and three researchers performed a screening of the titles and abstracts in triplicate. Articles containing pertinent data for this review’s objectives were selected for comprehensive screening. Systematic reviews and articles with data unrelated to the topics of interest were excluded. In case of doubt, the article was included for full-text screening. The full-text articles selected were reviewed to assess whether the review objectives were met. Gray literature, such as narrative reviews or conference abstracts, was excluded in a subsequent review. Each remaining article was reviewed further during the data extraction process. This step resulted in additional exclusions. In cases where articles contained similar results from the same data sets, the most recent one was included. 

The inclusion and exclusion criteria are listed in Table 1. One researcher (SMM) developed the data extraction log sheet for the eligible articles, which was reviewed by a second researcher (AB). A random review of 20% of the data extraction was then performed by two people (NO and SMM).

Using the PRISMA flowchart, of the 13,294 research articles identified through database searching, 187 were screened after removing duplicates (Figure 1). The second step of removing articles was excluded by physical checks (*n* = 11), the report sought for retrieval (*n* = 70), and reports that were not accessible for not having the full text (*n* = 38). The third step of screening was to exclude clinical trial results not reported (*n* = 7), very small study size (*n* = 1), or unrelated to the study objective (*n* = 14). This screening process yielded a total of 10 studies for final review.

### 2.3. Risk of Bias Assessment

Three investigators (SMM, NAW, AB) independently assessed the risk of bias in the included study against critical criteria: random sequence generation; allocation concealment; blinding of participants, personnel, and outcomes; incomplete outcome data; selective outcome reporting; and other sources of bias, in accordance with the methods recommended by The Cochrane Collaboration [7]. The following judgments were used: low risk, high risk, or unclear. Authors resolved disagreements by consensus and further article review if necessary.

## 3. Results

This meta-analysis included studies that measured the safety, efficacy, and immunogenicity of vaccinia (smallpox vaccine) as a vaccine for mpox in vaccinia-naïve vs. vaccinia-exposed groups in all studies (100%). Of the 10 studies included in the analysis, the majority (80%) were reported from the United States, 2 from Germany, and 1 from the U.S. and Mexico [9]. The age of participants ranged from 18 to 55 years. One study [10] represented Phase I, 7 Phase II, and 2 [11,12] Phase III clinical trials (Table 2).

### 3.1. Safety

In this analysis, a random effects model was used. Ten studies [9,10,11,12,13,14,15,16,17] were included in the pooled analysis of the safety of vaccinia (smallpox vaccine) as a vaccine for mpox in vaccinia-naïve vs. vaccinia-exposed. There were 4274 individuals allocated to vaccinia (vaccine-exposed group) and 3156 individuals allocated to control (vaccinia-naïve group). The pooled results suggest that the vaccinia naïve group had more side effects when compared to the vaccinia-exposed group (Odds ratio [OR] 1.92; 95% Confidence Intervals [CI]: 1.10–3.36; *p* = 0.02).

### 3.2. Efficacy

Fourteen pre-clinical and clinical trials demonstrated that ACAM2000^®^ had similar safety and immunogenicity to Dryvax^®^ [10,12,16]. However, the vaccines differed. ACAM2000^®^ is produced in cell culture from a clonal virus derivative of Dryvax^®^ and is considered a second-generation vaccine [16]. The primary goal of the study conducted by Overton et al. [15] was to determine the safety and immunogenicity of MVA-BN (also known as Imvamune, JYNNEOS, or Imvanex). It is a modified vaccinia Ankara-Bavarian Nordic smallpox vaccine, which was used in an immunocompromised subgroup for whom replication-competent smallpox vaccines are contraindicated. Their study demonstrated no difference in safety and tolerance of MVA-BN smallpox vaccine in HIV-infected subjects with CD4 counts as low as 200 cells/µ and in healthy individuals, regardless of their previous smallpox vaccination status. Furthermore, a vaccination with vaccinia virus (VACV)-specific immune response was observed due to MVA in HIV-infected subjects [15]. The study by Greenberg et al. [9] supports the efficacy of MVA against variola in vaccinia-experienced subjects. The study by Pittman et al. [12] demonstrated the efficacy of MVA-BN in vaccinia-specific Plaque Reduction Neutralization Test (PRNT) antibody response. It showed that vaccination prior to administration of ACAM2000^®^ results in an attenuated take. These study results support the US Food and Drug Administration’s (FDA) approval for the emergency use of MVA in the event of a smallpox outbreak.

### 3.3. Risk of Bias in Different Domains Observed in the Studies Analyzed

As presented in Figure 2, the domains assessed in this review were as follows: (a) Random Sequence Generation: It is a type of selection bias (biased allocation to interventions) due to inadequate generation of a randomized sequence. (b) Allocation Concealment: It is a selection bias (biased allocation to interventions) due to inadequate concealment of allocations prior to assignment. (c) Selective Reporting: This is a reporting bias due to selective outcome reporting. (d) Other Bias: This includes bias due to problems not covered elsewhere in the table. (e) Blinding of Participants and Personnel: This is a performance bias due to knowledge of the allocated interventions by participants and personnel during the study. (f) Blinding of Outcome Assessment: This is a detection bias due to knowledge of the allocated interventions by outcome assessors. (g) Incomplete Outcome Data: This is an attrition bias due to the amount, nature, or handling of incomplete outcome data.

The rating scales were High, Low, and Unclear. Where “Quote” is mentioned, there is a direct quote from the article reviewed. Where “Comment” is noted, it is the assessment or the author(s) for this review. All studies judged ‘unclear risk’ presented no identifiable risk for the corresponding domains. All other studies judged ‘low risk’, satisfying the corresponding domains’ requirements.

Greenberg et al., (2015) [9] was judged ‘high risk’ for the domain random sequence generation (selection bias). Comment: There was no description of the randomization process, and the domain selection reporting (reporting bias). Another study conducted by Greenberg et al. [9] was judged ‘high risk’ for the domain “Blinding of participants and personnel (performance bias)”. Quote: “The author acknowledged competing Interests: The study was funded by Bavarian Nordic through NIAID Contract Number N01-AI-40072. Rx Clinical Research, Inc. received grants from Bavarian Nordic. DVD is employed by Rx Clinical Research, Inc. The study investigators (DB, AK, RS, GV, NA, TM, DS, RN, PY, and PC) were employed by Bavarian Nordic at the time of this study.” Comment: Likely conflict of interests.

Overton et al., 2015 [15] was judged ‘high risk’ for domain allocation concealment (selection bias). Comment: The reason for considering the study “high risk” was because the study was conducted in a population at an increased risk of death or other life-threatening outcomes. Comment: no notation of DMC was mentioned. The report does not describe the allocation concealment process.

Overton et al., 2020 [13] was judged ‘high risk’ for the domain other bias. This is a direct quote from the article: “Declaration of Competing Interest. The authors declare the following financial interests/personal relationships which may be considered as potential competing interests”. Several authors (HW, DS, BK, GS, and KN) were employees and stakeholders of Bavarian Nordic GmbH (Biotechnology company in Planegg, Germany); PC, the company CEO. Investigators for this study (ETO, SJL, and JTS) were funded by Bavarian Nordic.

### 3.4. Bias of RCTs on Safety of Vaccinia

The studies have different risks of bias for different outcomes. The assessment results of the risk of bias are presented separately for each of the ten studies reviewed (Figure 3). Only five of the included studies had high risk among the seven domains. The Seaman study [10] had an overall high risk of bias.

### 3.5. The Forest Plot

Based on the pooled data, safety of the vaccinia vaccine goes in favor of the previously vaccinia-exposed group, compared with the vaccinia naïve group (OR 1.92; 95% CI of OR, 1.10 to 3.36; *p* = 0.02), There are moderate concerns of heterogeneity (I^2^ = 46%) and low risk of bias overall. The study conducted by Tack et al. [16] was the most significant one, having a total sample size of 753 and weight of 24%, and not crossing the line of null effect. Other studies that favored the vaccinia-exposed group were Overton et al., 2015 [15] (sample size, 579; weight, 23%), Overton et al., 2018 [11] (sample size 4005; weight 18.6%), and Overton et al., 2020 [13] (sample size, 87; weight, 2.7%). However, these samples crossed the vertical line of null effect, meaning they were not statistically significant. Two studies [11,16] and the overall effect favored the vaccinia-exposed group. Efficacy was determined by the whether or not an immune response was induced (noted in Table 2). This information was retrieved from the results of each clinical trial report. The overall result of the analysis of the 10 studies was statistically significant (Figure 4).

## 4. Discussion

Diagnostic and clinical details from the Halsell et al. study reported 18 serious adverse events (SAE) cases of myopericarditis following smallpox vaccination reported among 230,734 in the vaccinia naïve group [18]. There were no SAE cases among the previously vaccinated group (95,622) in their study. No pre-existing infectious etiologies that would predispose the hosts to myopericarditis were detected in serologic tests. All cases survived and returned to work. The researchers recommended screening for cardiovascular risk factors before vaccination. In a cohort study of Israel Defense Force recruits from 1991 to 1996, the authors [19] examined post-smallpox vaccination complications. The study reported that the complication rate increased with increasing primary vaccines. With a vast majority of cases seen in Europe, Australia, and America, the safety and immunogenicity of vaccinia for mpox are essential to determining how the modifications may affect an individual’s overall health and immunity [20].

Studies concluded that the smallpox vaccine efficacy ranges from three to five years of full protection [21]. It was also concluded that vaccinated subjects do not require booster vaccination because they have lifelong immunity to vaccinia once vaccinated [21]. In addition, immune responses can determine the nature of protection based on the closeness of the virus to orthopoxviruses. Poland and colleagues noted that the increase in mpox incidence rates since the eradication of smallpox vaccination is due to the prevalence of the immunologically naïve population [22]. With immunity waning over time, scientists must investigate the safety and efficacy of modified vaccinia against mpox and weaponized smallpox [21]. Existing literature is conflicted because researchers suggest that full protection is short-lived.

Poland et al., 2022 [22] have tested vaccines and antivirals, with a similar sequence between the genetic makeup of orthopoxviruses and the wide response of antibodies targeting more than 23 membrane and structural proteins. Their studies contribute to prior evidence of vaccina-specific immune responses, including animals such as chimpanzees. The United States has two vaccines to protect against MPX: JYNNEOS and ACAM2000. Vaccinia elicits antiviral protection and virus-neutralizing activity that remains intact for the patient’s life. [21]. JYNNEOS was approved in 2019 for both prevention of smallpox and mpox, while ACAM2000 is a second-generation vaccine adopted from a single clone isolate from Dryvax that reduced neurovirulence in animal models [22]. Though both are available, ACAM2000 is not recommended for the following individuals: immunocompromised individuals, patients having chronic diseases, pregnant women, and infants less than one year of age. In addition, the CDC recommends the use of JYNNEOS as the primary vaccine in preventing MPX due to its fewer potential side effects than ACAM2000 [6].

The JYNNEOS vaccinia modification offers a safer smallpox vaccination option for immunocompromised or other high-risk individuals who could not safely receive ACAM2000. The JYNNEOS clinical trial included 22 studies with a total of 7859 people, ages 18 through 80, who received at least one dose of the vaccine. The study of Greenberg et al. (2015) suggests that priority should be given to individuals born after 1972. In one clinical trial, researchers found that individuals with skin conditions who received JYNNEOS only experienced moderate skin reactions to the vaccine. Another study by Zitzmann-Roth et al. (2015) assessed the cardiac safety of individuals who received JYNNEOS. Scientists compared the placebo group to the vaccine group, and data did not show an increased risk of myocadiac or pericarditis after the JYNNEOS individuals with the controls [18].

### Limitations

It is not possible to know if our study has identified all relevant articles on the safety and efficacy of the vaccina or JYNNEOS vaccines. The electronic search produced 80% of the articles in the systematic review are from the U.S. clinical trials. The authors of this review only had access to aggregated data. Survival plots could not be constructed without having access to raw data, especially the “time” variable. The study focused on current modified vaccinia clinical trials that had published results. Although the overall summary results of 10 studies were statistically significant, because of the limited sample size of our study, we recommend further larger studies including clinical trial results from European and non-western countries.

## 5. Conclusions

The present study confirms differences in serious adverse events between individuals who were previously vaccinated with vaccinia and the vaccinia naïve individuals. Those previously exposed to vaccinia had fewer serious adverse events compared with the vaccinia naïve group. Vaccines have been designed to elicit an antibody response by B cells following inoculation with weakened or dead pathogens. There was not enough information about the efficacy and safety of smallpox vaccines for elderly populations. Information about the risk of mortality and morbidity associated with mpox vaccine development and administration among the elderly and immunocompromised individuals is critical to guide the U.S. vaccination and public health policies. It is conclusive that individuals previously exposed to vaccinia would have higher acquired immunity than the vaccinia naïve individuals. The data in the included clinical trials for this meta-analysis imply that the limited supply of vaccines can be more rationally applied to individuals who have never been vaccinated.

## Figures and Tables

**Figure 1 ijerph-20-02963-f001:**
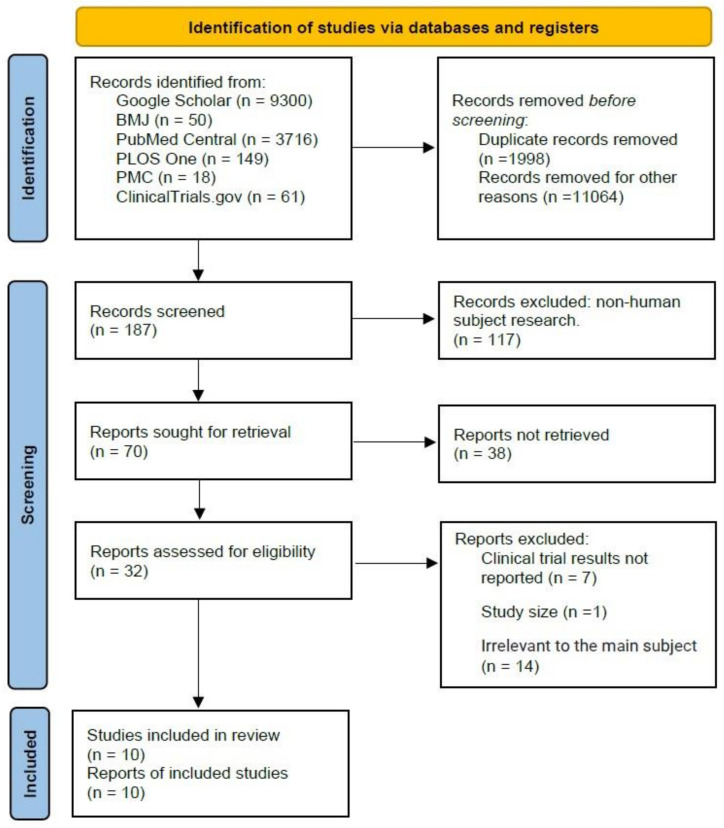
PRISMA flow diagram for inclusion and exclusion of studies.

**Figure 2 ijerph-20-02963-f002:**
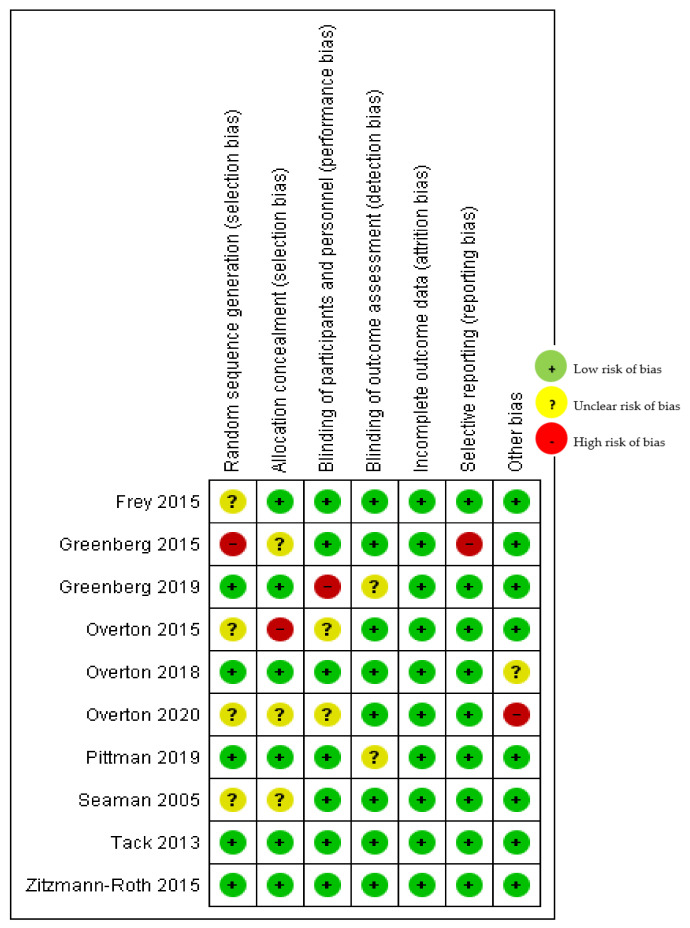
Risk of bias summary: review-authors’ judgements about each risk of bias item for the included study [9,10,11,12,13,14,15,16,17].

**Figure 3 ijerph-20-02963-f003:**
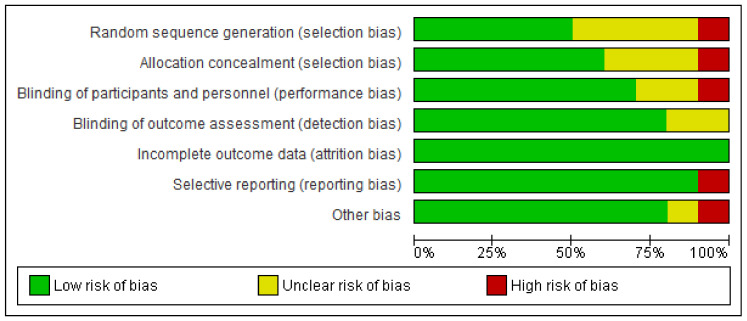
Risk of bias graph: review authors’ judgements about each risk of bias item presented as percentages across all included studies.

**Figure 4 ijerph-20-02963-f004:**
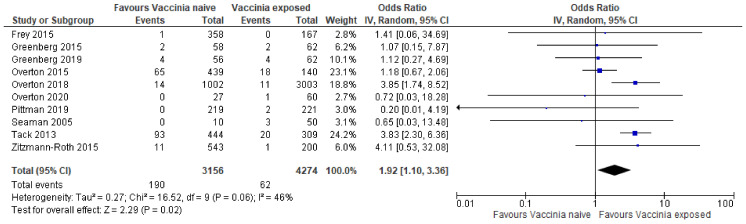
Forest Plot of Ten Studies [9,10,11,12,13,14,15,16,17].

**Table 1 ijerph-20-02963-t001:** Inclusion and exclusion criteria.

Inclusion Criteria	Exclusion Criteria
1. Human subjects in clinical trials	1. Non-human subjects
2. Published clinical trials	2. Publications prior to 2000
3. Studies for approved vaccines	3. Clinical trials without data published
4. Publications from 2000 to 2022	4. Sample sizes less than 20
	6. Clinical trials focusing on viral pathogenesis only
	7. Review articles

**Table 2 ijerph-20-02963-t002:** Study design, sample size, and outcome of the ten studies analyzed.

Author, Publication Year [Reference]	Study Design	Country	Age, y (Mean ± SD)	Total Sample	Immune Response Induced (Efficacy)	Outcomes
Overton et al., 2020 [13]	Phase II, randomized, double-blinded	USA	34.5 ± 6.61	87	Yes	Safety and immunogenicity
Frey et al., 2015 [14]	Phase II, randomized, triple blinded	USA	44 years	523	Yes	Safety and immunogenicity
Greenberg 2019 [9]	Phase II, non-randomized, open-label	USA and Mexico	27.9 ± 6.33	632	Yes	Safety and immunogenicity in people with atopic dermatitis
Greenburg 2015 [9]	Phase II, Randomized, Double-blind, Multicenter	USA	27.7 ± 6.28	651	Yes	Safety and immunogenicity
Overton et al., 2015 [15]	Phase II, Multicenter, Open-label, Controlled	USA	37.5 ± 8.0	579	Yes	Safety and immunogenicity
Overton et al., 2018 [11]	Randomized, Double-Blind, Placebo-Controlled Phase III Trial	USA	27.7 ± 6.3	4005	Yes	Immunogenicity, safety, and tolerability
Pittman et al., 2019 [12]	Phase III, Double-blind, Randomized, Dose-finding Study	USA	24.7 ± 4.2	440	Yes	Safety and immunogenicity
Seaman et al., 2010 [10]	Phase I/II, randomized, double-blinded, placebo-controlled	USA	25.2 ± 3.7	36	Yes	Safety and immunogenicity and surrogate efficacy (Dryvax challenge)
Tack et al., 2013 [16]	Partially Randomized, Partially Double-blind, Placebo-controlled Phase II Non-inferiority Study	Germany	29.8 ± 9.07	753	Yes	Safety and immunogenicity
Zitzman-Roth et al., 2015 [17]	Partially Randomized, Partially Double-blind, Placebo-controlled Phase II Non-inferiority Study	Germany	29.8 ± 9.07	745	Yes	Safety and immunogenicity

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
