# Peer review of "Safety and Efficacy of Post-Eradication Smallpox Vaccine as an Mpox Vaccine: A Systematic Review with Meta-Analysis"

_ijerph, 2023, doi:10.3390/ijerph20042963_

Round 1

Reviewer 1 Report

Thank the authors for the effort made. Allow me to make the following suggestions:

1. I recommend adding a summary table with the inclusion and exclusion criteria used.

2. There are paragraphs with different font sizes and not justified: points 3.2 and 3.3 for example.

3. Discussion, 1st paragraph, lines 222-223 “In the same study, those who were vaccine naïve were four times 223 more likely to die than the previously vaccinated group “: Can the authors make a graph using the Kaplan-Meier method?

Reviewer 2 Report

-In this manuscript, the authors (Malone, S. et. al.) present a systematic review and meta-analysis of the literature to determine the safety and efficacy of modified smallpox vaccines for treating mpox. Where the ‘Materials and Methods’ section detailed the literature selection strategy, data extraction, and bias assessments, methods corresponding to the safety analysis, efficacy analysis, and statistical analysis seem to be lacking. For example, in lines 215-216 the authors state that “the overall result of this study was statistically significant” however, no guidance on how that conclusion was produced was documented.

-Given the study only analyzed 10 total manuscripts, can the authors show that this sample size is enough to extract meaningful results for their meta-analysis?

-Figure captions are sparse and do not fully explain the extent of the data/image. For example, having and explaining a key for figure 2 would make it easier to interpret what the authors mean by the “?” without having to necessarily scan through the entire manuscript.

-Additionally, a couple of grammatical errors exist in the manuscript (ln. 149, 265), and an undefined acronym (ln. 197)
